# A Separated Receptor/Transducer Scheme as Strategy to Enhance the Gas Sensing Performance Using Hematite–Carbon Nanotube Composite

**DOI:** 10.3390/s19183915

**Published:** 2019-09-11

**Authors:** Nguyen Minh Hieu, Cao Van Phuoc, Truong Thi Hien, Nguyen Duc Chinh, Nguyen Duc Quang, Chunjoong Kim, Jong-Ryul Jeong, Dojin Kim

**Affiliations:** Department of Materials Science and Engineering, Chungnam National University, Daejeon 34134, Korea; cockyhieu@gmail.com (N.M.H.); caovanphuoc91@gmail.com (C.V.P.); anhientruong@gmail.com (T.T.H.); nguyenducchinh0605@gmail.com (N.D.C.); nguyenducquang0903@gmail.com (N.D.Q.); jrjeong@cnu.ac.kr (J.-R.J.)

**Keywords:** separated receptor and transducer, hematite–carbon nanotube composite, gas sensor

## Abstract

Nanocomposite structures, where the Fe, Fe_2_O_3_, or Ni_2_O_3_ nanoparticles with thin carbon layers are distributed among a single-wall carbon nanotube (SWCNT) network, are architectured using the co-arc discharge method. A synergistic effect between the nanoparticles and SWCNT is achieved with the composite structures, leading to the enhanced sensing response in ammonia detection. Thorough studies about the correlation between the electric properties and sensing performance confirm the independent operation of the receptor and transducer in the sensor structure by nanoparticles and SWCNT, respectively. Nanoparticles with a large specific surface area provide adsorption sites for the NH_3_ gas molecules, whereas hole carriers are supplied by the SWCNT to complete the chemisorption process. A new chemo-resistive sensor concept and its operating mechanism is proposed in our work. Furthermore, the separated receptor and transducer sensor scheme allows us more freedom in the design of sensor materials and structures, thereby enabling the design of high-performance gas sensors.

## 1. Introduction

Metal oxides have been extensively studied for chemoresistive gas sensors because of their high reactivity with various gas molecules as well as their adoptability to modern microfabrication technologies [1]. It is well known that the molecular adsorption of gas phases followed by the chemical reaction with the oxide occurs by the charge exchange between the gas molecule and oxide, and the consequent change of conductance change in the oxide is translated to the sensing signal through the metal electrodes underneath [2,3,4]. In such oxide gas sensors, the surface plays the role of the sensory receptor by the supply of gas molecular ionosorption sites while the bulk structure donates charges for the ionosorption reaction on the surface. At the same time, the bulk structure also works as a transducer, which reveals the change of conductance (or resistance) as an electric signal. A typical geometry for the operation of metal oxide gas sensors is configured by the receptor and transducer integrated into the single metal oxide body.

Since the sensor operating principle is strongly affected by the structure of the sensor, both ‘nanosize effect’ and ‘nanosize limit effect’ have been reported in conduction-type gas sensors. The nanosize effect is a well-known observation in chemoresistive gas sensors, which describes the increase in the sensing response as the dimension of the sensor structure decreases to a nanosize [4,5,6,7,8,9,10]. However, when the size decreases smaller than the depletion depth scale, the total depletion of carriers leads to the insulating nature in the sensor structure followed by the decrease of the sensor signal. Such declining response at the dimension below the depletion depth scale has been introduced as the nanosize limit [10,11]. Therefore, the highest response level can be achieved at the size comparable with that of the depletion depth for a given doping concentration of the oxide.

CNTs have been repeatedly reported to show the sensing capability toward NH_3_ at room temperature (RT, ~25 °C) [12,13,14,15,16,17]. The interaction of NH_3_ with the Fe and Fe_2_O_3_ at RT also was reported to occur via the dissociative molecular adsorption of NH_3_ on the Fe and Fe_2_O_3_ clusters [18,19,20,21,22,23,24,25,26,27], respectively. Hematite exhibited higher ammonia sensing performance at the elevated temperature (150 °C) than that at RT [26]. Meanwhile, the composite sensors, tin oxide-based carbon, showed a sensing capability toward NH_3_ [28,29,30] and NO_2_ [31]. Hieu et al. [28] showed that the SnO_2_/multi-wall CNT composite detected NH_3_ at RT. Lee et al. [29] reported about the SnO_2_/carbon nanoflake composite for the detection of NH_3_ at 400 °C. The different operating temperatures were possibly originated from the different mixing ratios between the carbon (a p-type conducting sensor operating at RT) and SnO_2_ (an n-type conducting sensor operating at high temperature). Rigoni et al. [30] also detected NH_3_ using the indium tin oxide (ITO) and SWCNT composite at RT. Interestingly, the conducting type of the composite sensor changed from the p-type semiconductor to the n-type with high NH_3_ adsorption due to the injection of many electrons into the structure.

In this study, we prepared composite sensors composed of iron or hematite (Fe_2_O_3_) and single-wall carbon nanotubes (SWCNTs), Fe:SWCNTs, and Fe_2_O_3_:SWCNTs, respectively, for the NH_3_ detection at RT, and scrutinized their sensing behaviours with pure hematite and SWCNT sensors. Fe:SWCNTs and Fe_2_O_3_:SWCNTs nanocomposite structures showed the enhanced NH_3_ sensing response at RT owing to the synergistic effect. In addition, based on the comparative study with single-body gas sensors, hematite or SWCNTs sensors, we explored the underlying sensing mechanism of the composite structure. The semiconductor depletion model for the nanosize effect and the nanosize limit effect was thoroughly revisited, thus we could overcome the nanosize limit via the new concept of separated receptors and transducers for the high-response chemoresistive gas sensors. While the large surface area of hematite nanoparticles with a thin carbon layer provides adsorption sites for ammonia gas molecules, the charges necessary for the ionosorption are supplied from the SWCNTs. The net result is the appearance of far more extended depletion in SWCNTs, enabling an enhanced sensor signal. We elaborate the origin of the synergy effect and the principle of the new sensor scheme. Furthermore, we showed that such a sensor scheme can be a general route to overcome the nanosize limit regardless of the junction types by comparison with the p-p junction composite of Ni_2_O_3_:SWCNT.

## 2. Experiment

### 2.1. Synthesis of the Fe_2_O_3_:CNT Structures

The Fe:CNT composites were fabricated on alumina substrates of 2.5 mm × 2.5 mm × 0.23 mm patterned with bar-type gold electrodes by a co-arc discharge method [11,32,33,34,35], which is commonly used to prepare nanocrystalline metal (or metal oxide)-CNT composites at a high temperature during the arc-discharge process. The distance between the Au electrodes was kept at 1 mm. In general, a co-arc discharge method produces the morphology of finely dispersed metal (or metal oxide) nanoparticles among the SWCNTs [32,34,35]. Firstly, the gold electrode-patterned Al_2_O_3_ substrates were ultrasonically cleaned sequentially with acetone, methanol, and deionized water for 15 min each, followed by N_2_ blow drying. The substrates were mounted on the inside wall of the arc-discharge chamber, as schematically shown in Appendix A. A hollow graphite tube with a length, outer diameter, and inner diameter of 160, 6.4, and 3 mm, respectively, was used as a carbon source (Appendix A). The hollow carbon tube was filled with different numbers of Fe wires. The arc-discharge process of the graphite tube filled with iron feedstock was performed for 20 min at an arc-discharge current density of 40 A cm^−2^ in a hydrogen atmosphere with a H_2_ partial pressure of 5.3 × 10^3^ Pa. The *as-deposited* sample is labelled by AD in Figure 1a that presents the flowchart of sample fabrication.

The AD sample, the Fe:CNTs composite, shows a highly porous structure, in which the Fe nanoparticles are uniformly dispersed in the entangled CNTs. It was observed that the iron particles are deposited by carbon during the arc-discharge process. Therefore, the Fe nanoparticles in the Fe:CNTs composite are actually the carbon-encapsulated Fe as can be seen in the TEM observation. When the AD sample was dipped into the methanol solution, the hydrophobicity of the CNTs rendered the porous Fe:CNT composite structure to be collapsed. Therefore, the *methanol-treated* (MT) sample with less porosity could be obtained as shown in Figure 1a [16]. Finally, the MT sample was *heat-treated* at 400 °C in air for 2 h to oxidize iron to iron oxide, thereby the Fe_2_O_3_:CNT composite was formed (sample HT). The carbon layer was still preserved on the surface of Fe_2_O_3_ even after heat treatment. 

In addition, pure CNT and pure iron oxide structures were prepared for the comparative study. The pure CNT (CNT-p) structures were prepared via the same arc-discharge method. However, the catalytic metals in CNTs were thoroughly removed by following the acid treatment as shown in Figure 1b. CNT-p showed a mat-like structure of the closely stacked SWCNTs morphology. The pure hematite structure (H-p) was fabricated through a chemical method using an iron solution in dimethylformamide (DMF, (CH_3_)_2_NC(O)H), followed by the heat treatment at 500 °C (Figure 1c). 

### 2.2. Morphology and Structure Characterization

Most characterizations and measurements were performed from the samples with 20 min of the arc-discharge process, which showed the highest sensing performance. The morphologies of the fabricated structures were investigated by field-emission scanning electron microscopy (FE-SEM, JSM 700F, JEOL, Tokyo, Japan) and transmission electron microscopy (TEM, JEM-ARM200F, JEOL, Tokyo, Japan). The crystalline structures and chemical binding natures were examined by X-ray diffraction (XRD, X’pert PROMPD, PANalytical, Amelo, The Netherlands) with Cu Kα radiation and Raman spectrometry (Horiba Jobin Yvon, LabRAM HF-800, Horiba, Paris, France), respectively.

### 2.3. Sensing Property Measurements

The gas sensing properties were measured using a Picoammeter/Voltage source (Keithley 6487, Keithley Instruments, Solon, Ohio, USA) in the homemade measurement system [34]. Appendix A displays the schematic of the sensor measurement system equipped with the gas flow vacuum chamber, mass flow control (MFC) system, substrate heater, and computer. Sensor structures were degassed in the vacuum chamber at 300 °C before the sensing measurement to remove any preadsorbed water molecules. The target gases such as NH_3_, H_2_S, H_2_, CH_4_, etc. were supplied by gas cylinders and diluted up to 1000 ppm by nitrogen. The gases were further diluted in dry air by varying the gas concentration at a constant dry air flow rate of 100 sccm when fed into the test chamber. The gas concentrations were determined by $C\left(\mathrm{ppm}\right)=C_{\mathrm{std}}\left(\mathrm{ppm}\right)\times f/\left(f+F\right)$, [36,37] where f and F are the flow rates of the analyte gas and the carrier gas, respectively, and C_std_ (ppm) is the concentration of the analyte gas in the gas cylinder. C_std_(ppm) was 1000 ppm balanced with nitrogen for all of the gases. Dry air was used as a carrier gas, and the gas flow rate was controlled by a mass flow controller. The resistances were measured by applying 1 V bias between the two electrodes with flowing air (R_o_) or the target gas (R_g_) into the chamber. 

## 3. Results and Discussion

The sample AD (the Fe:CNT composite) showed a highly porous morphology with large volume among the SWCNTs as shown in the SEM image (Figure 2a). The methanol treatment led the porous composite structure to collapse and form a compact mat-like structure (sample MT) due to the hydrophobicity of the SWCNTs (Figure 2b) [38,39]. The sample HT (the Fe_2_O_3_:CNT composite) was obtained by thermal oxidation of the MT sample, in which Fe nanoparticles were oxidized to Fe_2_O_3_ nanoparticles while preserving the porosity of the MT sample as well as the thin carbon layer on the surface (Figure 2c). Transmission electron microscopy (TEM) images clearly presents the dispersed hematite particles on the SWCNT network in the HT sample (Figure 2d). The entangled SWCNT bundles with the diameter of ~10 nm supported Fe_2_O_3_ nanoparticles, of which the size range is large from several to ~30 nm. The high-resolution TEM image with the electron diffraction pattern of the inset (Figure 2e) confirms the crystalline Fe_2_O_3_ nanoparticles in the Fe_2_O_3_:CNT composite structure. The hematite particles were encapsulated by amorphous or graphitic carbon layers as shown by Figure 2e,f. The carbon layers encapsulating the Fe and Fe_2_O_3_ nanoparticles were preserved during the oxidation process at 400 °C.

The XRD patterns of the AD and HT samples are shown in Figure 3a. Those of pure Fe_2_O_3_ (H-p) and SWCNT (CNT-p) are displayed together for the comparison. The diffraction peak at ~44° matches (400) of Fe in the AD sample, indicating the distribution of Fe nanoparticles on SWCNTs. Meanwhile, diffraction peaks corresponding to the polycrystalline hematite (Fe_2_O_3_; JCPDS 00-024-0072) and graphite (JCPDS 98-002-0123) were identified in the HT sample. The Raman spectra of the composites are shown in Figure 3b. The G- and D-mode peaks at ~1600 cm^−1^ and ~1400 cm^−1^ represent the graphitic lattice and the defective graphite structure, respectively, therefore the intensity ratio of G to D peaks can be considered a measure of the crystalline characteristic of the single-wall CNT [40,41]. All composites revealed the similar intensity ratio of G to D peaks indicating the preserved graphitic character in SWCNT during the process. This indicates that the crystallinity of the CNT in the composites was not noticeably degraded after the methanol or heat treatment, as exhibited by the sharp G-mode peak with respect to the D-mode peak.

The electrical properties of the composite structures are measured as shown in Figure 3c. All composite samples exhibited excellent ohmic contacts with the Au electrodes at RT in air as indicative by the linearity in the current versus voltage curves. The high resistance of ~3.1 × 10^5^ Ohm of the as-deposited Fe:CNT composite (AD) originates from the high porosity of the structure. The high porosity in the entangled nanowires comprises relatively a small number of contacts among the CNTs, which leads to the high resistance by a small cross-section area for the current path. The other Fe:CNT composite after the methanol treatment, the MT sample, shows two orders of magnitude lowered resistance of ~6.8 × 10^3^ Ohm. The methanol treatment led to the collapse of the porous structure followed by formation of the compact mat-like structure [38,39], in which the cross-section area for the current path is large due to the great increase in the number of contacts among the CNTs. HT reveals the same compact structure with MT, however, Fe nanoparticles were oxidized to semiconducting oxide particles, Fe_2_O_3_. Despite similar compact structures between MT and HT, the much higher resistance of Fe_2_O_3_ than Fe caused a far increase in the resistance, ~1.3 × 10^4^ Ohm. A partial burn-out of the SWCNTs could contribute to the increase of the resistance in the HT structure [38].

The resistance changes of the HT composite toward 500 ppm NH_3_ gas diluted in dry air were measured at different temperatures, as shown in Figure 4a. The increase of the resistance upon exposure to the reducing gas like NH_3_ indicates that the HT composite is a p-type sensor, in which the positive charge carrier hole (h^+^) is supplied to the adsorbed NH_3_ molecules, thereby enabling ionosorption (e.g., ${\mathrm{NH}}_{3}+h^{+}\to {\mathrm{NH}}_{3}^{+}{}_{\mathrm{ad}}$). The change of the resistance was highest at RT, however it decreased by the increase of the temperature. The similar p-type semiconducting sensing behaviours and the temperature dependence were observed in the Fe:SWCNT composite (AD and MT samples) as shown in Figure 4b. The RT sensing response was attributed to the molecular adsorption of NH_3_ on the sensor (not combustion of NH_3_). Furthermore, the p-type behaviours in the sensing mechanism indicates that p-type SWCNTs are the main contributor for the charge exchange during NH_3_ adsorption.

The RT response-recovery curves of the composite structures (AD, MT, and HT) are presented in Figure 5a. The response and recovery times of the HT structure were ~1.2 and ~1.8 min, respectively, as estimated in previous studies [8,9]. The measurements from pure SWCNT (CNT-p) and pure hematite (H-p) structures with 500 ppm NH_3_ are also plotted for comparison. The corresponding resistance changes of the sensor structures are presented in Figure 5b. It can be immediately noted that the H-p sensor with very high resistance showed negligible response and the CNT-p sensor with very low resistance also showed a small response to NH_3_ at RT. However, the composite-type sensors revealed the enhanced chemoresistive response level. The observed sensing response with the composites is elucidated by the synergistic interplay between the hole carrier concentration in the transducer and the adsorption sites density of the receptor. Holes are supplied by p-type SWCNTs, of which concentration is directly translated to the sensor resistance. The density of adsorption sites depends on the physical and chemical nature of the nanoparticles distributed on SWCNTs. We thoroughly investigate the synergetic effect in the sensing response by examination of the electronic and morphological properties of the structures.

Ionosorption is chemical adsorption (chemisorption) of the gas molecule on the material surface as an ionic state, which can be completed by either capture or release of electrons by or from the material, respectively. Therefore, a molecule impinging on the receptor can end up with adsorption followed by the charge exchange with the transducer. The charge supplied to or extracted from the transducer leads to a resistance change of the transducer. The ratio of resistances before and after gas adsorption (R_o_ and R_g_, respectively) is taken as the sensor response signal, which is defined by Equation (1),
(1)$$S=\frac{R_{g}}{R_{o}},$$
for sensing of oxidizing gases on n-type doped materials or reducing gases on p-type materials. The reverse ($S=R_{o}/R_{g}$) can be defined for sensing oxidizing gases on p-type doped materials or reducing gases on n-type materials. 

The simple condition, the uniform thin film oxide sensor with a thickness (t), is illustrated in Figure 6a. The resistance is inversely proportional to the thickness, $R_{o}={\mathrm{At}}^{-1}$ (A is arbitrary). If a reducing gas adsorbs by the hole supply from the p-type film (such as the NH_3_ adsorption on SWCNT), the surface region becomes depleted of holes with the depletion depth (d) from the each side, leading to the change of the film resistance, $R_{g}=A{\left(t-2d\right)}^{-1}\;$ [36]. Therefore, the response of the thin film sensor is given by Equation (2),
(2)$$S=\frac{t}{t-2d}.$$

S approaches the unity (= 1) or negligible response in the condition of t >> d (Figure 6a) [10]. The nano-size effect in gas sensing is observed by the enhancement of response as thickness (t) approaches twice of the depletion depth, i.e., t = ~2d, of which the condition is illustrated in Figure 6b. The nano-size effect is highest when t = ~2d or the film thickness is comparable with the depletion depth in the given doping concentration. Since the depletion in the transducer occurs by the molecular adsorption of the gas phase on the receptor surface and consequent charge extraction, the total depletion condition is given by $n_{d}t=\sim N_{\mathrm{ad}}$, where n_d_ and N_ad_ are the carrier concentration of the transducer (cm^−3^) and the areal density of molecular adsorption on the receptor (cm^−2^), respectively, in the steady state for a given gas concentration. Considering the resistance and the adsorption site intensity in the structure, the difference in the sensing response among AD, MT, and HT (see room temperature data in Figure 4b) can be elucidated by the sensor structural conditions between Figure 6a,b. 

However, in the case of t < 2d, the total charges in the transducer structure are smaller than the steady-state adsorption molecules (n_d_t < N_ad_). Only part of the steady-state adsorption sites can be occupied. The completely depleted state in the transducer before and after the gas adsorption is depicted in Figure 6c, where the absence of the sensing response, i.e., S = ~1, is detected. It should be noted that Equation (2) cannot be applied to this condition. The maximum response as proposed in Figure 6b begins to decline as the thickness decreases like the schematic in Figure 6c. We already called this the declining response with the decrease in the sensor dimension below the depletion depth scale as ‘nanosize effect limit’. We had repeatedly observed the nanosize effect limit in several oxide gas sensors [9,10,11,42]. We have to note that both of the nanosize effect (increase in the sensing response with decreasing the sensor size towards the depletion depth scale) and the nanosize limit (decrease in the sensing response with further decreasing the sensor size below the critical limit of the depletion depth) are conceived from the independence between the receptor and transducer action in the single-body scheme of the gas sensor. Therefore, simultaneous control of the dimensions of the oxide element, carrier concentration, and adsorption site density is required to achieve the maximum response signal in the conventional single-body oxide sensors.

To discuss the synergetic effect of the composite sensors, we need to understand the origin of zero and negligible response levels from the H-p and CNT-p sensors, respectively, as shown in Figure 5a. Detection of NH_3_ in CNT-based sensors at RT has been reported [12,13,14,15,16,17], and the relatively good recovery at RT was available due to the weak adsorption binding between CNT and NH_3_ [43]. However, the CNT-p exhibited a small response level (S = ~1.02 or R_g_ = ~R_o_ in Equation (1)), indicating that the resistance modulation in the SWCNT by the NH_3_ adsorption is very small. The small response is attributed to the relatively high hole carrier concentration (i.e., high conductance in SWCNTs) compared with the small adsorption site density of the SWCNTs (n_d_t >> N_ad_). This case corresponds to the condition schemed in Figure 6a, where the adsorption sites are occupied at the steady state, however, the consequent surface deletion depth is relatively small. 

Meanwhile, the origin of the no response (S = ~0) in the H-p sensor (Figure 5a) is different with that in the CNT-p sensor. It is reported that an n-type hematite sensor responded to ammonia at a high temperature like other oxide gas sensors [26]. However, the hematite film exhibited an agglomeration among nanometric hematite particles, thereby revealing a very high resistance (>GΩ) due to fully depleted electron carriers as shown in Figure 5b. The semi-insulating hematite film corresponds the condition of n_ox_t << N_ad_ as presented in Figure 6c. Ionosorption of 500 ppm NH_3_ molecules followed by the release of electrons into the hematite could not lead to any detectable resistance change in the H-p sensor, as manifested by the nearly zero response. 

Contrary to the poor sensing responses of the SWCNT and hematite, the enhanced response in their composite structures can be understood by investigation about their electronic and morphological structures. Namely, the resistance of the Fe:SWCNT composite is comprised by competition between the high-resistance Schottky contact between the Fe and SWCNT and low-resistance ohmic contact among the SWCNTs. Therefore, the resistances of the composites are located in between that of CNT-p and H-p (Figure 5b), which is directly reflected to the operating condition. The sensor operating conditions of the composites lie between the conditions presented in Figure 6a,c. Meanwhile, the proposed morphology of the composite sensors with the enhanced sensing response is depicted in Figure 6d. The high-performance composite structure can be designed by consideration about two aspects. (1) The carbon-encapsulated Fe (or Fe_2_O_3_) nanoparticles with great surface area work as receptors for the NH_3_ adsorption, and (2) the nanoparticle/SWCNT heterojunctions decrease the conductance of the SWCNT transducer. Both effects cooperatively push to the condition of Figure 6b of higher response. 

The effect of concentration and distribution of Fe_2_O_3_ nanoparticles on the sensor response of composites were examined as well. The Fe_2_O_3_ concentration in the Fe_2_O_3_:SWCNT composites could be increased by adding more Fe wires into the graphite rod during the co-arc-discharge process. The synthesis using the more Fe wires produced the composite with the higher carbon-encapsulated Fe_2_O_3_ concentration. The response to 500 ppm NH_3_ of the structures are shown in Figure 7a. The higher responses were obtained with the more Fe wires due to higher receptor concentration in the composites. Since the NH_3_ adsorb on SWCNT as shown by the response of the CNT-p sensor, the high distribution of carbon-encapsulated hematite particles will increase the specific adsorption sites for NH_3_.

Whereas, when Fe was sputter deposited on the mat-like SWCNTs thin film, a thin film of Fe seats on the SWCNT mat. The following oxidation process converts Fe to Fe_2_O_3_ nanoparticles agglomeration without the carbon shell layer. The sputter-based composite will show a morphology of thin film Fe_2_O_3_ nanoparticle receptor piled up on the mat-like SWCNT transducer. The cross-section views of the structures are shown in Appendix A that signifies increase in the thickness with the sputtering time. The sputter-based composite also revealed higher response with the Fe_2_O_3_ thickness (Figure 7b). It was already repeatedly shown that the NH_3_ molecules can adsorb on Fe or Fe_2_O_3_ nanoparticles at RT via the molecular adsorption reaction ${\mathrm{NH}}_{3}\to {\mathrm{NH}}_{3}^{+}{}_{\mathrm{ad}}+e^{-}$ or the dissociative adsorption reaction ${\mathrm{NH}}_{3}\to {\mathrm{NH}}_{2}{}_{\mathrm{ad}}+H^{+}{}_{\mathrm{ad}}+e^{-}$, etc. [19,20,21]. In this way, either the ammonia adsorption on carbon-encapsulated Fe_2_O_3_ or the direct adsorption on Fe_2_O_3_ observed in Figure 7 can be supported. 

The microscopic view about the ammonia adsorption on the carbon-encapsulated hematite nanoparticle is presented in Figure 8. The Fe_2_O_3_/SWCNT heterojunction results in the formation of the hole depletion region in the SWCNT. When the NH_3_ molecules adsorb on the carbon-encapsulated Fe_2_O_3_ receptor at RT, electrons released via the adsorption reaction are transferred to the SWCNTs to recombine with holes. The electron transfer can be achieved by either surface conduction or tunneling through the hematite particles. The resultant increase in the hole depletion depth in the SWCNT transducer is translated to the sensor signal following Equation (2). From this perspective, an enhanced sensor response, i.e., far increased resistance modulation of the SWCNT transducer, is available by increase in the adsorption site density for the NH_3_ molecules. The higher adsorption can be achieved by the higher concentration of nanoparticle receptors. 

We also fabricated the Ni_2_O_3_:SWCNT as the p-type-oxide:SWCNT composite structure by the same method used for the preparation of the Fe_2_O_3_:SWCNT composite sensor. The temperature dependence of sensing responses to 100 ppm NH_3_ is presented in Appendix A. All results follow those from the Fe_2_O_3_:SWCNT composite. This indicates that the enhanced SWCNT sensor performance can be achieved by adding any type of receptor if it can provide adsorption sites for NH_3_. 

The various ammonia sensing properties of the HT sensor (Fe_2_O_3_:CNT composite) at RT are summarized in Figure 9. In the test of the repeatability of sensing, the baseline resistance after recovery continued to increase by the repetition of the sensing cycles as shown in Figure 9a. The chemical forms of ammonia adsorption can be variable in reality [19,20,21], and a finite amount of $H^{+}$ and/or $H_{2}^{+}$, which binds more strongly with the adsorption sites than ${\mathrm{NH}}_{3}^{+}$, remains adsorbed during the recovery cycle [19]. The remnant adsorption can be accumulating as the response-recovery cycles are repeated. The linearity of the HT sensor at RT for ammonia detection was measured at the various concentrations in the range from 50 to 500 ppm as shown in Figure 9b. The responses of the HT sample to other reducing and oxidizing gases were also examined, which shows high selectivity of the sensor to ammonia at RT (Figure 9c). The preliminary examination shows that the composite structure can be further developed to commercial sensor structures. It is also interesting to note in Figure 9c that the composite structure can be used as a H_2_S sensor at high temperatures with good selectivity.

The humidity in the air is considered one of the most critical issues to operate the oxide gas sensor at RT. The H_2_O molecules strongly affect the sensing signals via adsorption on the surface of the sensor. At low temperatures below 400 K, preferentially dissociated water molecules form the chemisorption layer of hydroxyl ions on the very surface, and further water molecular impingement forms multilayers of physisorbed water molecules [44,45,46,47]. The water molecular adsorption provides the ionic conduction path, thereby increasing the conductance of the oxide. In addition, electrons are released via the water molecular adsorption [3], which also contributes to the further increase of the conductance in the case of n-type semiconducting oxides. We found that the n-type ZnO thin film can be used as a selective humidity sensor because it shows the conductance increase by relative humidity (RH) (Appendix A). Rapid decrease of the sensing response above ~60% RH may account for an enhanced conduction due to several layers of the physisorbed water molecules. The conductivity change resulting from the physisorbed water molecules as well as the simultaneous electron injection into the oxide can manifest the humidity sensing of the ZnO.

Whereas, the behaviour of the p-type Fe_2_O_3_:CNT composite sensor is opposite with ZnO as shown in Figure 9d, in which the conductance of the sensor was decreased by the humidity (see the curve with the zero NH_3_ flow). Since SWCNTs also respond to water molecules [48], the opposite humidity effect on the sensor conductance indicates that a part of the holes in the SWCNTs are compensated by the released electrons from the chemisorbed water molecule. Figure 9d shows that the NH_3_ gas in a ppm concentration range competitively adsorb on the surface in the humidity as indicative by sensing signals. Note that the ionic conduction by the adsorption of water molecules is not significant. Therefore, the ammonia sensing signal of HT in the humid condition still can result from the hole conduction mechanism in the SWCNT transducer. As observed, the ammonia gas concentration cannot be precisely determined in humidity unless the humidity is measured separately. The simplest solution is the monitoring of the humidity at the same location. Many semiconductor type humidity sensors have been developed to be mounted together with the chemoresistive sensor [45]. In this case, a microprocessor to calibrate the signals based on the humidity is required as well. 

## 4. Conclusions

An ammonia sensing response was thoroughly studied in the Fe:SWCNT, Fe_2_O_3_:SWCNT, and Ni_2_O_3_:SWCNT nanocomposite structures, in which carbon-encapsulated metals or metal oxide nanoparticles are finely distributed among the SWCNT bundles. The much higher sensing response of the composites than that of the pure hematite and SWCNT structures could be obtained by the greater charge modulation in the sensor structure. Such condition could be motivated by the increase in the number of molecules adsorbed on nanoparticles that formed the intimate contact with SWCNTs. The distribution of fine nanoparticles of Fe, Fe_2_O_3_, and Ni_2_O_3_ in the composites increases the specific adsorption sites for NH_3_. In turn, the more adsorption of NH_3_ molecules accordingly leads to the wider depletion depth in the SWCNT structure. The synergetic effect that enhanced the sensing performance was elaborated by our new sensor scheme, in which the physically separated but electrically connected nanoparticles and SWCNTs function simultaneously as receptors and transducers, respectively, during the sensor operation. This understanding allows us to propose the new concept of a separated receptor and transducer scheme in the conduction-type gas sensors, which enable the optimization of the receptor and the transducer as independent materials and structures. Our study provides the guidance to choose the best receptor and transducer materials toward the development of the high-performance sensor that cannot be achieved in the existing design scheme. 

## Figures and Tables

**Figure 1 sensors-19-03915-f001:**
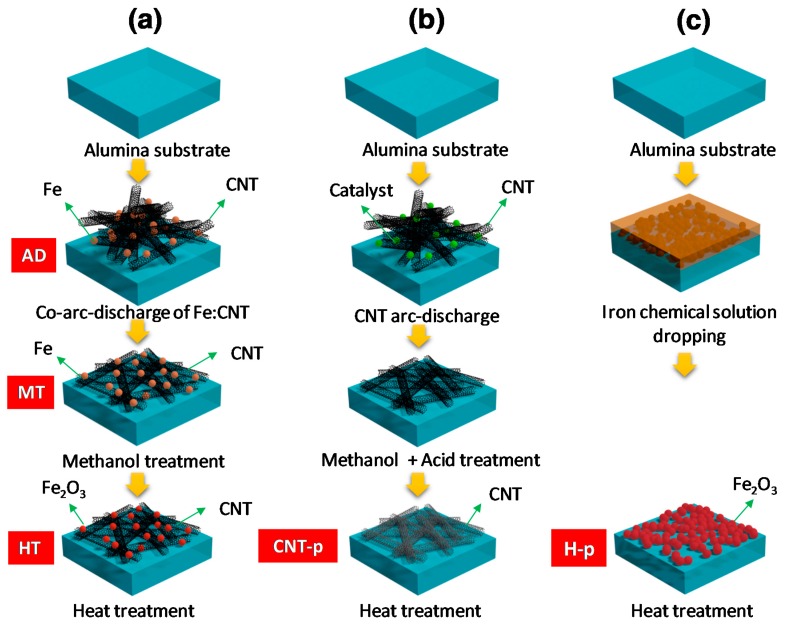
Experimental procedure of fabricating the (**a**) as-deposited (AD) and methanol-treated (MT) iron:carbon nanotubes (Fe:CNT) and heat-treated (HT) hematite (Fe_2_O_3_):CNT composite structures, (**b**) pure CNT, and (**c**) pure Fe_2_O_3_.

**Figure 2 sensors-19-03915-f002:**
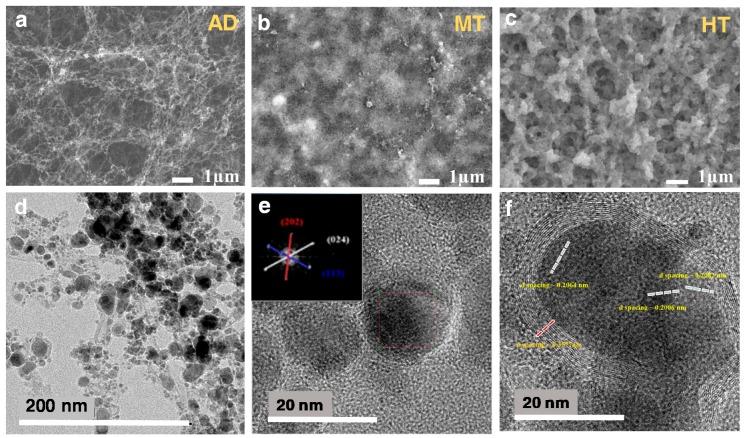
Top-view SEM images of the (**a**) AD, (**b**) MT, and (**c**) HT samples. (**d**) The TEM image of HT showing hematite nanoparticles entangled with the single-wall carbon nanotube (SWCNT) network. High-resolution (HR)TEM images confirms the existence of (**e**) amorphous and (**f**) graphitic carbon layers on the surface of hematite particles.

**Figure 3 sensors-19-03915-f003:**
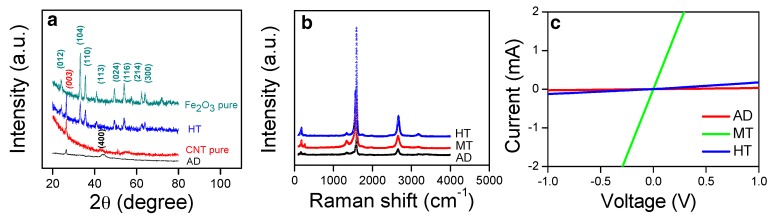
(**a**) XRD patterns of the HT and AD samples along with the pure CNT and Fe_2_O_3_ samples. The peaks confirmed Fe, polycrystalline hematite, and graphite. (**b**) Raman spectra of the AD, MT, and HT samples. (**c**) Current versus voltage curves of the AD, MT, and HT composite structures.

**Figure 4 sensors-19-03915-f004:**
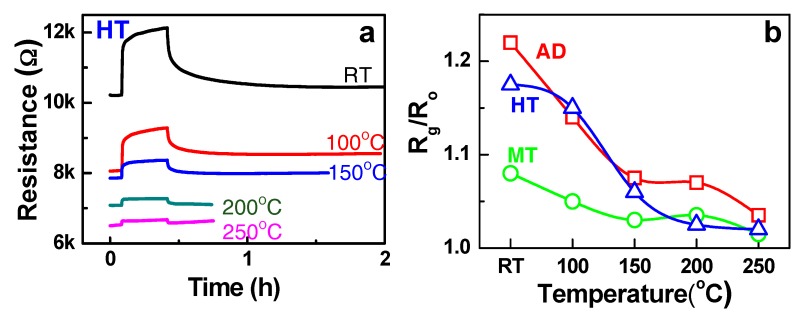
(**a**) The response-recovery curves of the HT structure measured for 500 ppm NH_3_ with varying temperatures. (**b**) The temperature dependence of the response levels of the AD, MT, and HT (500 ppm NH_3_) structures.

**Figure 5 sensors-19-03915-f005:**
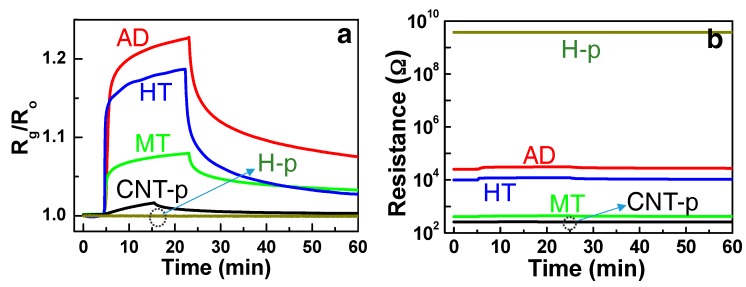
(**a**) Sensor response-recovery characteristics of AD, MT, and HT structures to 500 ppm ammonia at room temperature (RT). The responses of the composites showed a synergetic effect compared with pure hematite (H-p) and pure SWCNT (CNT-p). (**b**) The same presented by the change in resistances.

**Figure 6 sensors-19-03915-f006:**
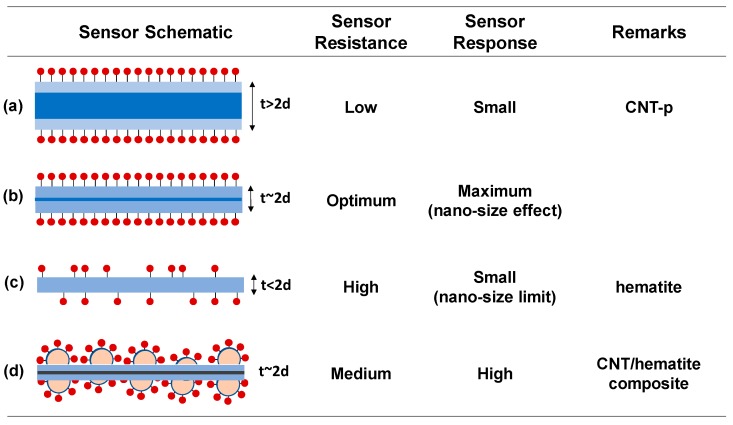
The schematic showing nanostructured sensor conditions with different performance. (**a**) The structural dimension of the sensor material is far greater than the depletion depth. The response is small due to too high conductance by the great conduction region. (**b**) The dimension of the structure approaches the depletion depth, which is the condition of the maximum nano-size effect in sensing. (**c**) The dimension of the structure is smaller than the depletion depth. Too high resistance leads to the negligible modulation of the resistance followed by the small response during chemisorption on the sensor. Note the surface adsorption sites are all occupied in the condition of (a) and (b), but are partially occupied in (c) due to the limited charge carriers in the structure. (**d**) The increase in both charge carriers and adsorption site density in optimized composite structures can reveal the high sensing capability like the condition of (b).

**Figure 7 sensors-19-03915-f007:**
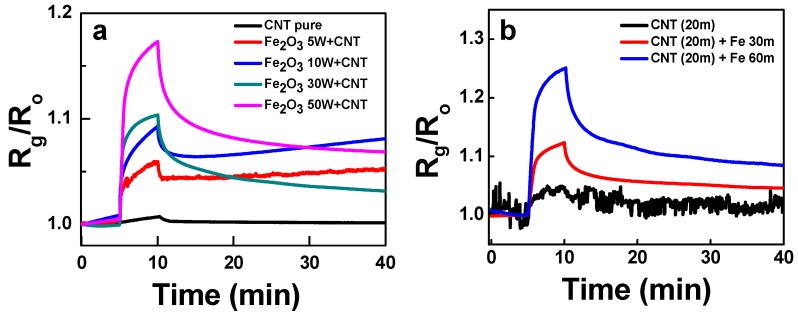
The response to 500 ppm NH_3_ of the Fe_2_O_3_:SWCNT composite structures (**a**) fabricated by the arc-discharge method with varying number of iron wires inserted to the graphite rod from five to 50 wires, and (**b**) fabricated by sputter deposition of Fe for 30 and 60 min on the SWCNT mat structures.

**Figure 8 sensors-19-03915-f008:**
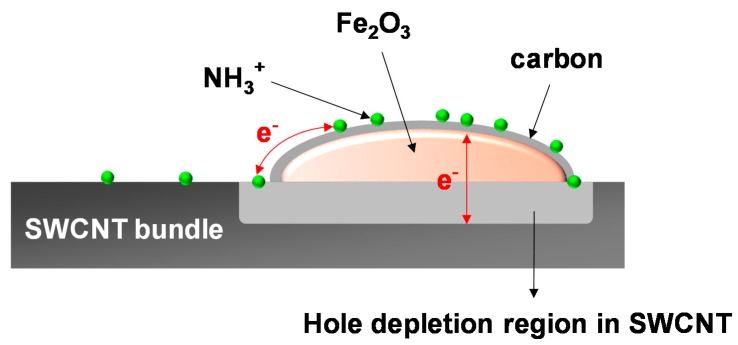
Schematic of the carbon-encapsulated Fe_2_O_3_ nanoparticle in contact with the SWCNTs bundle, forming the hole depletion region. The electrons released via adsorption of the NH_3_ molecules on the carbon-encapsulated Fe_2_O_3_ transport to SWCNTs by surface conduction or tunneling. The electrons recombine with holes in SWCNTs, which expands the depletion region at the Fe_2_O_3_/SWCNT heterojunction.

**Figure 9 sensors-19-03915-f009:**
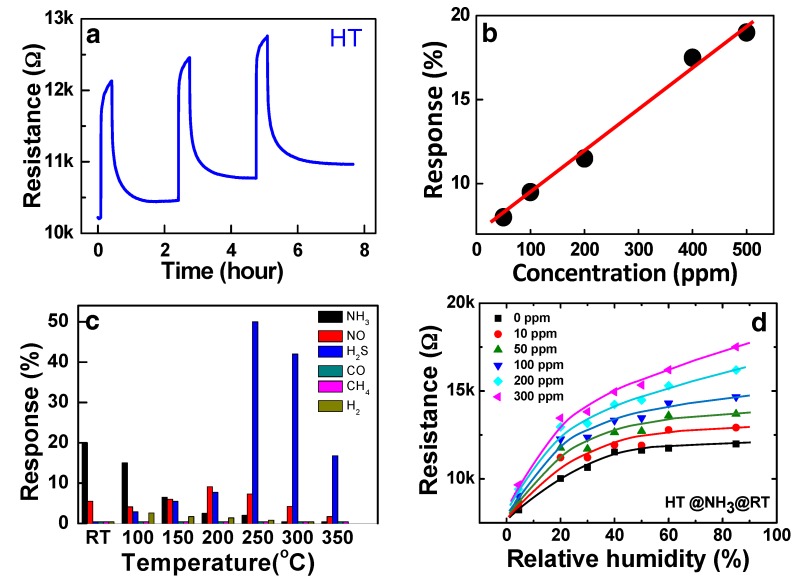
(**a**) Sensing reproducibility of the HT sample showing good repeatability but with increasing baseline resistance for 500 ppm NH_3_ concentration at RT or ~25 °C. (**b**) Linearity of HT measured in the concentration range between 50 and 500 ppm NH_3_ at RT. (**c**) Gas selectivity of HT tested at various temperatures for each of the 500 ppm concentrations. (**d**) Humidity dependent resistance of HT measured at RT under varying NH_3_ concentration and RH.

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
