# Peer review of "A Separated Receptor/Transducer Scheme as Strategy to Enhance the Gas Sensing Performance Using Hematite–Carbon Nanotube Composite"

_sensors, 2019, doi:10.3390/s19183915_

Round 1

Reviewer 1 Report

The paper deals with the amonia gas sensing performance of carbon nanotube - hematite composite structure. Results are comapred to clean metals and CNT - Fe composite. 

The intriduction part is fine however some more 'fresh' refferences may ba added because most of the references are older than 3 years. 

The experimental part is disrobed well.

The results and discussion part is well organised and includes deep discussion on the sensing mechanisms of obtained structures. However in my opinion the used formulas like in eq. 2 in the page 8 need some references. 

In my opinion the paper is well organized, novel and interesting for reader in the filed. The topic of the article mach to the Sensors journal. I recommend this paper for publication after minor revision according to abovementioned comments and minor point bellow.   

The minor points: 
- in page 4 the resistance unit should be given as Ω or eventually Ohm not 'ohm'.
- in the page 3 at the end of paragraph 2.1. there is a problem with used fount/style. 

At the end of paper the authors contribution should be given. 

-

Author Response

Q1: However in my opinion the used formulas like in eq. 2 in the page 8 need some references. 

A1: Thank you for your comments, following your recommendation, we added reference for that equation and numbered as 37.

Q2: In page 4 the resistance unit should be given as Ω or eventually Ohm not 'ohm'.

A2: Thank you for your comments, following your recommendation, we fixed all the “ohm” to “Ohm”

Q3:  In the page 3 at the end of paragraph 2.1. there is a problem with used fount/style. 

A3: Thank you for your comments, following your recommendation, we fixed the problem.

Q3:  At the end of paper the authors contribution should be given. 

A4: Thank you for your comments, following your recommendation, we added the detailed author contribution at the acknowledgement part.

Reviewer 2 Report

This article entitled “a separated receptor/transducer scheme as strategy to enhance the gas sensing performance using hematite/carbon nanotube composite” described a method to fabricate hematite-CNT composite with comprehensive chemical and physical characterizations. In addition, the concept of the separated receptor and transducer-based chemo-resistive mechanism and sensor structures was investigated, to support the synergetic gas sensing performance. The manuscript is in a concise and clear format, which is qualified for publication.

Author Response

 Thank you very much for the positive review.

Reviewer 3 Report

                                The major issue with   ammonia sensors is to discriminate water vapor and ammonia.

This paper also fails to distinguish water vapor from ammonia; in other words, there is significant interference from water vapor.

The following are my comments.

1.       “The holes necessary for NH3 adsorption are supplied by the SWCNT. While a new chemo-resistive sensor concept and mechanism is proposed for the synergetic gas sensing performance, the separated receptor and transducer sensor scheme allows us more freedom in the design of sensor materials and structures”

What is the meaning of holes? For a reader it may be a vacant space.

2, Interestingly, these authors observed that the conducting type of the composite sensor changed from p-type to n-type with high NH3 adsorption due to the injection of many electrons into the structure.

Change p-type semiconductor to n-type

3.       The co-arc-discharge method leads to the morphology that shows finely dispersed metal or metal oxide nanoparticles among the SWCNTs.

The sentence is not clear.

4.       Figure 2 C Why is thermally oxidized sample and as prepared sample have high resistance?

The resistance of Fe and Fe2O3 are nearly the same. One is a conductor and the other is a semiconductor.

5.       FIGURE 4 B SHOWS THE REPOSNE TO Ammonia of the three samples. Why there is difference at room temperature?

6.       Figures 4 and 5 are contradicting values for the same concentration of ammonia.

7.       S approaches the unity or negligible response in the condition of t >> d (Figure 6a), which can occur with thick sensors of low doping or thin sensors of high doping

Suitable reference.

8.       Figure 7 shows the electron is travelling to full the hole. How was it concluded? The electron affinity of Fe2O3  should be considerably higher than CNT. If not can this be supported by suitable data.

9.       Figure 1 is confusing. AD, MT and HT should follow the arrows.

The following references should be considered,

Korean Journal of Materials Research Vol.26 No.4 pp.187-193

Electrochimica Acta, Volume 180, 20 October 2015, Pages 1059-1067

The paper may be considered for publication after major revision.

Author Response

Reply to Reviews

Thank you very much for reviewing the manuscript with helpful comments. Itemized answers are made under the best understanding of the questions and comments.

Reviewer 3

Q1 : “The holes necessary for NH3 adsorption are supplied by the SWCNT. While a new chemo-resistive sensor concept and mechanism is proposed for the synergetic gas sensing performance, the separated receptor and transducer sensor scheme allows us more freedom in the design of sensor materials and structures”

What is the meaning of holes? For a reader it may be a vacant space.

A1: Hole is defined by the majority charge carrier in p-type doped semiconductor like SWCNT. It was clearly defined in p. 11 to avoid any confusion.

‘….the positive charge carrier hole (h+) is supplied to the adsorbed NH3 molecules, thereby leading to the ionosorption (eg, NH3+h+→NH3+ad ).’

Q2:  Interestingly, these authors observed that the conducting type of the composite sensor changed from p-type to n-type with high NH3 adsorption due to the injection of many electrons into the structure.

Change p-type semiconductor to n-type

A2: We added ‘semiconductor’ to make the sentence clearer per suggestion.

Q3 :  The co-arc-discharge method leads to the morphology that shows finely dispersed metal or metal oxide nanoparticles among the SWCNTs.

The sentence is not clear.

A3: We modified the sentence with references to “In general, the co-arc-discharge method produces the morphology of finely dispersed metal (or metal oxide) nanoparticles among the SWCNTs [32,34,35].”

Q4: Figure 2 C Why is thermally oxidized sample and as prepared sample have high resistance?

The resistance of Fe and Fe2O3 are nearly the same. One is a conductor and the other is a semiconductor.

A4: Thank you for your comment. We described in more detail for the readers about the change in the resistances in relation with the morphology in p. 8.

‘The high resistance of ~3.1×105 Ohm of the as-deposited Fe:CNT composite (AD) originates from the high porosity of the structure. The high porosity in the entangled nanowires means relatively a smaller number of contacts among the CNTs, and consequently, a smaller cross-section area for the current path leading to the higher resistance. However, the Fe:CNT composite (MT) structure was made by methanol treatment of AD structure. The methanol treatment leads to collapse of the porous structure to form a compact mat structure [36,37] in which the number of contacts among the CNTs has greatly increased resulting in far increased cross-section area for the current path. The net result is the two orders of magnitude lowered resistance of ~6.8×103 Ohm. HT has the same compact structure as MT but the Fe particles have changed to semiconducting oxide particles, Fe2O3. Since Fe or Fe2O3 nanoparticles are dispersed in the CNT mat forming a part of the current path, the much higher resistance of Fe2O3 than Fe led to a far increased resistance of ~1.3×104 Ohm.’

Q5:  FIGURE 4 B SHOWS THE REPOSNE TO Ammonia of the three samples. Why there is difference at room temperature?

A5: The result originates from the interplay between the sensor geometry (surface adsorption site intensity) and the sensor conductance as explained with Fig. 6. This is the main background for the proposal of separated receptor/transducer in this paper. For the question raised, a note is added in p. 13 as ‘Note that the difference in the response between AD, MT, and HT (see room temperature data in Fig. 4b) can be also explained by the sensor structural conditions sitting between Fig. 6a and 6b in relation with the given resistance and the adsorption site intensity in the structure.’

Q6: Figures 4 and 5 are contradicting values for the same concentration of ammonia.

A6: We plotted from the same data and do not see any contradiction or differences. The resistance of HT is ~104 ohm in Fig. 4a and 5b, the room temp response for AD, HT, MT are the same in Fig. 4b and 5a.

Q7:  S approaches the unity or negligible response in the condition of t >> d (Figure 6a), which can occur with thick sensors of low doping or thin sensors of high doping

Suitable reference.

A7: We removed the latter added description to avoid unnecessary confusion. Instead our related paper was referenced. The part is edited to ‘S approaches the unity or negligible response in the condition of t >> d (Figure 6a) [10]’

Anyway, the intended of the previous description was; For low-doped case d is relatively high and t must be accordingly farther thicker for t>>d. However, in highly doped case, d is smaller and t does not have to be so thick as in the low doped case for t>>d condition.

Q8: Figure 7 shows the electron is travelling to full the hole. How was it concluded? The electron affinity of Fe2O3  should be considerably higher than CNT. If not can this be supported by suitable data.

A8: The electron affinity determines the electron transport before two materials come into contact (more precisely, Fermi energy determines the electron transport direction). Once two materials have contacted and establish equilibrium, the electron transport due to Fermi energy difference results in ‘contact potential’, which is equivalent to the junction shown in Fig. 7. We need to consult semiconductor junction theory for full description.

Once a junction is formed as in Fig. 7, the electron transport thereafter in the system is governed by the Coulomb’s law. The arrow indicates this transport. If NH3 adsorbs in a form of NH3+, a hole should be supplied from somewhere (or an electron has to move into the material). The transport path may be either the surface transport or tunneling.

Q9: Figure 1 is confusing. AD, MT and HT should follow the arrows.

A9: The samples are produced at the step-by-step progress stage of the fabrication process. Description in p. 6 was revised for clearer explanation.

Q10: The following references should be considered,

Korean Journal of Materials Research Vol.26 No.4 pp.187-193

Electrochimica Acta, Volume 180, 20 October 2015, Pages 1059-1067

A10: We added the first reference.

Reviewer 4 Report

The manuscript is dealing with (NH3) gas sensing of oxide nanoparticles (Fe2O3 & Ni2O3) and SWCNTs composites deposited onto alumina substrate with bar-type Au electrodes. The composites were deposited by so-called co-arc-discharge method and treated by different methods (densification by methanol treatment & heat treatment in air) after the deposition. Authors performed basic characterization of the composite and reference samples after these treatment stages by X-ray diffraction, Raman spectroscopy, and scanning/transmission electron microscopies. To understand the sensing mechanism of the composite sensors authors investigated their electrical response to NH3 gas at different temperatures, NH3 concentrations and after different treatment steps as well as at different humidity levels. Authors claim separated receptor – transducer approach to enhanced the gas sensor performance.

The topic of the manuscript is fitting to the journal scope and it is potentially interesting for specific reader community but at this level of text preparation I do not recommend it to be accepted for publication in the journal. Serious changes are needed to be done because the manuscript contains imprecise formulation and speculative statements without taking into account other possible explanations of observed effects. Additionally gas sensing performance is not well discussed and the composite-electrode gas response is not eliminated to strengthen the significance of the separated receptor/transducer strategy.

Here are some examples of imprecise (or even false) formulations:

In 3. Results and Discussion:

Authors: …” due to the oxidation of iron nanoparticles and partial thermal decomposition of the CNTs” 

Nanotubes are thermally very stable (no thermal decomposition at 400°C) but they can be oxidize at used temperature.

Using of word “defective carbon“ or “hexagonal carbons” etc is incorrect (it is  sp2 hybridized carbon and defective graphene (or graphite) structure,…).

Authors statement: “… the absolute content of the SWCNTs drastically decreased … . … This result indicates lose and burning out of the CNTs after methanol and heat treatments…)”

There is NO reason for decrease of SWCNTs content after methanol treatment (Raman intensity depends on more parameters not only on amount of CNTs) of course heat treatment on air could be responsible for decrease of CNT content.

Authors: …”We elaborate the synergetic response by the electronic/morphological properties of the sensor structures”.

There are not really morphological information available: What is the distance between electrodes? What is diameter and length distribution of CNTs? What is density of the composite after different treatment steps? How thick is Fe film deposited in the case of evaporated iron?

It will be interesting to know what is C:Fe ratio (w or at) too.

Discussion about depletion depth in SWCNT is questionable and should be make more precise. SWCNT has only 1 layer what is then depletion depth? (Fig 7?). In CNT-composites there is enough space for gas to penetrate into the “film” and therefore it is questionable to discuss depletion depth from the “film” surface. Be clear if you mean depletion depth in the sense of length -distance from the nanoparticle on the same tube.

Authors: “…NH3 molecules adsorb on the Fe atomic sites ….” and “ … less ammonia could be adsorbed on the Fe sites … “, and “ … the open surface of Fe … “

There is no open surface of Fe nanoparticle. There is either Fe encapsulated/covered by carbon material or there is iron oxide (iron oxidize on air even on Tr).

To the discussion about temperature effect- authors should add experiments at lower T (Tr) after high T (250°C)  - Fig 4. Is the sensor response reproducible? Which effect is responsible for possible irreproducibility caused by higher temperature?

To characterize effect of Au-CNT interface on gas sensor response authors should add the experiment with passivated electrode areas – so only CNT composite between electrodes will be sensing part of the sensor.

The discussion about response time should be broadened taking into account another possibilities as mentioned in text.

For the statement that nanoparticles “generate defects on the SWCNT surface” need some support discussion. No TEM no local Raman experiments supporting this statement are presented.

… there are missing informations CNT-p. Which catalyst have been used for CNT synthesis, What is Differenet in comparison to co-arc-dischage method, …

In conclusion:

Authors: …. “the composite type gas sensor structureas in our study … achieve the optimum sensor performance”.

What are optimum sensor performance? What is sensitivity, detection limit, response rate, recovery time, reproducibility, selectivity of response, … What is composite thickness, density, CNT diameter distribution, nanoparticle size distribution and concentration in respect to CNTs?

Author Response

Reply to Reviews

Thank you very much for reviewing the manuscript with helpful comments. Itemized answers are made under the best understanding of the questions and comments.

Reviewer 4:

Q1: Due to the oxidation of iron nanoparticles and partial thermal decomposition of the CNTs” 

Nanotubes are thermally very stable (no thermal decomposition at 400°C) but they can be oxidize at used temperature.

Using of word “defective carbon“ or “hexagonal carbons” etc is incorrect (it is  sp2 hybridized carbon and defective graphene (or graphite) structure,…).

A1: Thank you for your comments, we edited through the text following your recommendation.

Q2: Authors statement: “… the absolute content of the SWCNTs drastically decreased … . … This result indicates lose and burning out of the CNTs after methanol and heat treatments…)”

There is NO reason for decrease of SWCNTs content after methanol treatment (Raman intensity depends on more parameters not only on amount of CNTs) of course heat treatment on air could be responsible for decrease of CNT content.

A2: Following the suggestion the ambiguous discussion part was removed.

Q3: Authors: …”We elaborate the synergetic response by the electronic/morphological properties of the sensor structures”.

There are not really morphological information available: What is the distance between electrodes? What is diameter and length distribution of CNTs? What is density of the composite after different treatment steps? How thick is Fe film deposited in the case of evaporated iron?

It will be interesting to know what is C:Fe ratio (w or at) too.

A3: The distance between the electrodes is 1 mm (as added in p. 5). The SWCNT bundle showed ~10 nm diameter in Fig. 3a (as added description in p. 9), and the length should be variable but cannot be measured. The density of composite (by which the reviewer may ask about the Fe:C ratio) was not examined, but the reasoning of Q2 as the reviewer raised above was assumed. Through the process steps, Fe content will be maintained unless the irreproducibility in the fabrication. We did not use evaporation for Fe film.

Q4: Discussion about depletion depth in SWCNT is questionable and should be make more precise. SWCNT has only 1 layer what is then depletion depth? (Fig 7?). In CNT-composites there is enough space for gas to penetrate into the “film” and therefore it is questionable to discuss depletion depth from the “film” surface. Be clear if you mean depletion depth in the sense of length -distance from the nanoparticle on the same tube.

A4: The same question has also occurred to me. The SWCNT resolved by SEM is actually SWCNT ‘bundle’ and which is the SWCNT scale contacting with Fe or Fe2O3 in Fig. 7. Therefore, the ~10 nm bundle is taken as the ‘film’ to form the junction with Fe or Fe2O3.

►It is beyond the scope of this paper to discuss the ‘hole depletion’ in the monolayer SWCNT (it is atomic scale) because depletion is a concept derived for bulk electronic materials while the question is about atomic scale charge behavior. Nevertheless, many researchers for the time being adopt the bulk semiconductor theory to explain the physics in nano geometries. Justification or debate for this interpretation requires further study and is beyond the scope of this study.

Q5: Authors: “…NH3 molecules adsorb on the Fe atomic sites ….” and “ … less ammonia could be adsorbed on the Fe sites … “, and “ … the open surface of Fe … “

There is no open surface of Fe nanoparticle. There is either Fe encapsulated/covered by carbon material or there is iron oxide (iron oxidize on air even on Tr).

A5: We removed ‘open’ because most of the Fe particles observed by many TEM are encapsulated by various thickness of material (carbon-like but not identified).

►However, it is also true that NH3 adsorption is related to Fe (or Fe2O3) not carbon layer that encapsulating Fe (or Fe2O3) because CNT itself shows low adsorption to NH3. Therefore, we logically derived the conclusion that NH3 adsorption is related to Fe not the encapsulating carbon. In this sense, the physically observed carbon encapsulation is not relevant to the NH3 adsorption, and we can say NH3 adsorb on Fe sites. Many references cited in the paper claimed that Fe atoms react with NH3 for adsorption-desorption.

Q6: To the discussion about temperature effect- authors should add experiments at lower T (Tr) after high T (250°C)  - Fig 4. Is the sensor response reproducible? Which effect is responsible for possible irreproducibility caused by higher temperature?

A6: We routinely degas the sensor by heating up to 300oC before sensing property measurements at RT in order to get rid of the humidity effect. This was added in p. 7.

Thus measured reproducibility at RT shows good reproducibility in Fig.8 (a).

Q7: To characterize effect of Au-CNT interface on gas sensor response authors should add the experiment with passivated electrode areas – so only CNT composite between electrodes will be sensing part of the sensor.

A7: Thank you for your comments. Definitely the Au/CNT [or more precisely Au/(CNT+Fe) and Au/(CNT+Fe2O3)] contacts can affect the sensing results, and avoiding the possibility will more strongly support the results and discussion. This kind of dispute can always happen with all the similar sensor structures.

First, we did not do that because the passivation itself provides different kind of sensing material and we again have to prove the passivation material does not affect the results. Second, the argument is based on the assumption that the Au/CNT contact is the major sensing mechanism which has to be justified. If then it will be another paper. As we observed, the sensor conductance has changed with the nature of the CNT/FE(or Fe2O3) composite, not with the Au/CNT contact resistance. We assume the contact resistance is relatively smaller compared to the sensor body resistance via the ohmicity measurements in Fig. 2d.

Q8. The discussion about response time should be broadened taking into account another possibilities as mentioned in text.

For the statement that nanoparticles “generate defects on the SWCNT surface” need some support discussion. No TEM no local Raman experiments supporting this statement are presented.

A8: Firstly, the low energy supply at RT will generally delay the gas-sensor reaction kinetics. This is well known as we also repeatedly observed. The estimated minute-scale response time could be originated from the room temperature environment. However, secondly, the gas diffusion to the defects at the CNT/Fe2O3 interface can further delay the response and recovery (If defects are really generated there and they act as the adsorption sites for NH3).

It is generally accepted that any contact between materials of different structures forms discontinuity of the lattice at the interface, which is nothing but the defects. The defects used in the paper is the ‘generic defects’ whose nature is not known. Note that even if we provide the existence of some defects, it cannot still be a direct support they are the adsorption sites. The do in situ measurements for the direct proof and will be another study.

Q9:  there are missing informations CNT-p. Which catalyst have been used for CNT synthesis, What is Differenet in comparison to co-arc-dischage method, …

A9: We revised the part in more detail in p. 7, which should be read as ‘The pure CNT (CNT-p) structures were prepared via the same arc-discharge method followed by acid treatment, which removed the catalytic metals leaving pure CNTs behind, as shown in Fig. 1b.’ For the fabrication of pure hematite was also revised to ‘Pure hematite (H-p) was also fabricated through a chemical method with an iron solution with dimethylformamide (DMF, (CH3)2NC(O)H), followed by heating up to 500oC to oxidize Fe (Fig. 1c).’.

Q10: Authors: …. “the composite type gas sensor structure as in our study … achieve the optimum sensor performance”.

What are optimum sensor performance? What is sensitivity, detection limit, response rate, recovery time, reproducibility, selectivity of response, … What is composite thickness, density, CNT diamater distribution, nanoparticle size distribution and concentration in respect to CNTs?

A10: We agree that all the above key parameters need to be provided for the final sensor development. In this paper, however, we are reporting the newly observed synergy effect in the response signal and focused on understanding of the underlying mechanism, which is necessary for further continued study towards the engineering sensor development. We think our study of RT sensor has just begun. Many of the issue have to be solved for the engineering sensors specification stated above. They are remained as future studies. This point was added in conclusion as ‘Therefore, we pursued this strategy through the composite type gas sensor structures and consistently explain the enhanced sensing responses in the structures.’

Round 2

Reviewer 4 Report

As mentioned in the previous referee report, the manuscript should be improved prior publication. The manuscript still contains imprecise formulation and speculative statements and authors did not improved the manuscript seriously after the first referee feedback.

Here are some examples of repeatedly imprecise (or even false) formulations:

As I commented recently there is no open surface (for adsorption of NH3) of Fe nanoparticle because there are no uncovered Fe nanoparticles at ambient conditions. There are either Fe particles encapsulated/covered by carbon material or there are iron oxide particles (iron oxidize on air even on Tr)!  Authors could not write:

page17/ Authors: “When NH3 molecules impinge on Fe or Fe2O3 receptor particles at RT”

and

page 23/ Authors: “Nanoparticles of Fe, Fe2O3, and Ni2O3 supply adsorption sites for NH3, and their adhesion to the SWCNTs also generate defects on the SWCNT surface as the adsorption sites.”

Referee: … because, in general, there are no Fe particles accessible to NH3 on air performed experiments. Authors should rewrite these (and similar) statements.

Page 18/ Authors:  “The loading of Fe2O3 nanoparticles could be increased using the other technique, in which Fe was sputter deposited on the SWCNT mat that is a collapsed SWCNT structure by methanol treatment and then heat-treated (Fig. S3). As a result, Fe2O3 nanoparticles are deposited above the SWCNT network rather than intermixed among the SWCNTs like the HT structure. The increase in the Fe2O3 nanoparticle thickness of the sputter-based composite also revealed higher response signals than the SWCNT mat. This behaviour confirms the role of the hematite as the supply of adsorption sites for NH3 molecules. “

Referee: Text above (without referencing other info source) indicates that authors performed additional experiment(s) with deposited iron on SWCNT mat.

 I asked previously: How thick is Fe film deposited in the case of evaporated iron?

With explanation of authors: “We did not use evaporation for Fe film.”

Referee: I am sorry, I should ask:  How thick is Fe film deposited in the case of iron sputtering onto SWCNT mat?  If the experiment was not performed (sputtering of Fe onto SWCNT mat) authors should reformulate the text mentioned above because they show only experiment where Fe particles were co-deposited with SWCNTs during arc-dischage synthesis of SWCNTs.

Another example of imprecise formulation is using the term “good relaxation in the recovery” is very unspecific and the sentences should be changed (page 15, page 17).

Another example of imprecise formulation is the sentence from page 16/authors: “However, the hematite film fabricated using the co-arc-discharge method exhibited an agglomeration among nanometric hematite particles (Fig. 3a), thereby showing a very high resistance (> GΩ) as shown in Fig. 5b due to fully depleted …“

Referee: There are no hematite films fabricated by co-arc-discharge method presented in this manuscript. There are iron oxide - SWCNT composites fabricated by co-arc-discharge method. The sentence should be removed or reformulated.

To be more rigorous (and in agreement with authors response ) authors should change the inserted text in Fig 7 to SWCNTs or SWNT film as the hole depletion region (thickness)  in the illustration is uncorrect for individual SWCNT.

Referee: As I mentioned there are speculative statements without taking into account (in discussion) other possible explanations of observed effects (plus imprecise formulation):

For example: page 16/Authors: “Then, the resistance increased again in HT compared with MT, which results from the conversion of Fe to Fe2O3, as already explained in Fig. 2d.”

Referee: How authors know that difference in resistance is not caused by convertion of metallic CNTs to semiconducting CNTs during oxidation at 300°C in air or even by destroing small diameter tubes (different ratio of RBM modes in Raman spectrum for HT and MT sample in Fig 3c) or by different doping or strain level (shifted 2D mode in Fig 3c) or another effect? They should “soften” the statement if they cannot exclude another impacts. … Plus … the statement: “ as already explained in Fig 2d” is incorrect. The measurement did not explain the “Fe to Fe2O3 conversion” … maybe observed measurement could be explain by the Fe to Fe2O3 conversion.

etc.

Author Response

Reply to Reviews

Thank you very much for reviewing the manuscript with helpful comments. Itemized answers are made under the best understanding of the questions and comments.

Reviewer 4:

As mentioned in the previous referee report, the manuscript should be improved prior publication. The manuscript still contains imprecise formulation and speculative statements and authors did not improved the manuscript seriously after the first referee feedback.

Here are some examples of repeatedly imprecise (or even false) formulations:

As I commented recently there is no open surface (for adsorption of NH3) of Fe nanoparticle because there are no uncovered Fe nanoparticles at ambient conditions. There are either Fe particles encapsulated/covered by carbon material or there are iron oxide particles (iron oxidize on air even on Tr)!  Authors could not write:

page17/ Authors: “When NH3 molecules impinge on Fe or Fe2O3 receptor particles at RT”

and

page 23/ Authors: “Nanoparticles of Fe, Fe2O3, and Ni2O3 supply adsorption sites for NH3, and their adhesion to the SWCNTs also generate defects on the SWCNT surface as the adsorption sites.”

Referee: … because, in general, there are no Fe particles accessible to NH3 on air performed experiments. Authors should rewrite these (and similar) statements.

Page 18/ Authors:  “The loading of Fe2O3 nanoparticles could be increased using the other technique, in which Fe was sputter deposited on the SWCNT mat that is a collapsed SWCNT structure by methanol treatment and then heat-treated (Fig. S3). As a result, Fe2O3 nanoparticles are deposited above the SWCNT network rather than intermixed among the SWCNTs like the HT structure. The increase in the Fe2O3 nanoparticle thickness of the sputter-based composite also revealed higher response signals than the SWCNT mat. This behaviour confirms the role of the hematite as the supply of adsorption sites for NH3 molecules. “

Referee: Text above (without referencing other info source) indicates that authors performed additional experiment(s) with deposited iron on SWCNT mat.

 I asked previously: How thick is Fe film deposited in the case of evaporated iron?

With explanation of authors: “We did not use evaporation for Fe film.”

Referee: I am sorry, I should ask:  How thick is Fe film deposited in the case of iron sputtering onto SWCNT mat?  If the experiment was not performed (sputtering of Fe onto SWCNT mat) authors should reformulate the text mentioned above because they show only experiment where Fe particles were co-deposited with SWCNTs during arc-dischage synthesis of SWCNTs.

Thank you so much for the comments. Carbon-encapsulated hematite and sputter deposited hematite on SWCNT are related topics, and thus, we respond here together.

Although HRTEM is a highly selective method to identify an existence, a few pictures showed that the Fe2O3 particles synthesized by the co-arc-discharge method are coated by amorphous and/or graphitic layers. Therefore, the fundamental receptor material structure was changed to carbon-encapsulated particles, and accordingly revised throughout the whole context. A new HRTEM picture was added in Fig. 2 to support the revision, and the related discussion for the sensing mechanism was consistently revised with modified Fig. 8 (previously Fig. 7). The naming in the schematic of Fig. 8 was also accordingly modified.

Fig. 7 was newly added or moved from the supporting information. Here both methods for increasing the nanoparticles (co-arc-discharge and sputter deposition) are introduced to show both carbon-encapsulated hematite and pure hematite receptors can adsorb NH3. Fig. S3 of SEM pictures were added to show the thickness change with the sputter deposition time.

Another example of imprecise formulation is using the term “good relaxation in the recovery” is very unspecific and the sentences should be changed (page 15, page 17).

The ambiguous statement was removed.

Another example of imprecise formulation is the sentence from page 16/authors: “However, the hematite film fabricated using the co-arc-discharge method exhibited an agglomeration among nanometric hematite particles (Fig. 3a), thereby showing a very high resistance (> GΩ) as shown in Fig. 5b due to fully depleted …“

Referee: There are no hematite films fabricated by co-arc-discharge method presented in this manuscript. There are iron oxide - SWCNT composites fabricated by co-arc-discharge method. The sentence should be removed or reformulated.

To be more rigorous (and in agreement with authors response ) authors should change the inserted text in Fig 7 to SWCNTs or SWNT film as the hole depletion region (thickness)  in the illustration is uncorrect for individual SWCNT.

We are so sorry. The description for hematite film fabrication is not relevant to the sample in this study. The description was corrected.

Fig. 7 (here Fig. 8) legend was fully revised including the term ‘SWCNT bundle’.

Referee: As I mentioned there are speculative statements without taking into account (in discussion) other possible explanations of observed effects (plus imprecise formulation):

For example: page 16/Authors: “Then, the resistance increased again in HT compared with MT, which results from the conversion of Fe to Fe2O3, as already explained in Fig. 2d.”

Referee: How authors know that difference in resistance is not caused by convertion of metallic CNTs to semiconducting CNTs during oxidation at 300°C in air or even by destroing small diameter tubes (different ratio of RBM modes in Raman spectrum for HT and MT sample in Fig 3c) or by different doping or strain level (shifted 2D mode in Fig 3c) or another effect? They should “soften” the statement if they cannot exclude another impacts. … Plus … the statement: “ as already explained in Fig 2d” is incorrect. The measurement did not explain the “Fe to Fe2O3 conversion” … maybe observed measurement could be explain by the Fe to Fe2O3 conversion.

etc.

The main reason we proposed for the increased resistance with HT from MT was the conversion of Fe to Fe2O3 (metallic to semiconducting), not from metallic CNT to semiconducting CNT. Furthermore, therefore, the doping, strain, etc related to the CNT property cannot be discussed either. However, a slight burn-out of the CNT during the oxidation process at 400oC could increase the resistance, and this possibility was states in p. 10 with reference. We showed the SWCNT just begins to burn-out from 400oC, which is the temperature we used for oxidation of Fe. The related experimental part was also revised for more clear explanation of the processing.
